# *Brassica napus* Plants Gain Improved Salt-Stress Tolerance and Increased Storage Oil Biosynthesis by Interfering with CRL3^BPM^ Activities

**DOI:** 10.3390/plants12051085

**Published:** 2023-03-01

**Authors:** Emily Corbridge, Alexandra MacGregor, Raed Al-Saharin, Matthew G. Garneau, Samuel Smalley, Sutton Mooney, Sanja Roje, Philip D. Bates, Hanjo Hellmann

**Affiliations:** 1School of Biological Sciences, Washington State University, Pullman, WA 99164, USA; 2Department of Applied Biology, Tafila Technical University, Tafila 66110, Jordan; 3Institute of Biological Chemistry, Washington State University, Pullman, WA 99164, USA

**Keywords:** BPM, *Brassica*, cullin, CRL3, E3 ligase, fatty acids, PEST, proteasome, storage, stress, yield, canola

## Abstract

Generating new strategies to improve plant performance and yield in crop plants becomes increasingly relevant with ongoing and predicted global climate changes. E3 ligases that function as key regulators within the ubiquitin proteasome pathway often are involved in abiotic stress responses, development, and metabolism in plants. The aim of this research was to transiently downregulate an E3 ligase that uses BTB/POZ-MATH proteins as substrate adaptors in a tissue-specific manner. Interfering with the E3 ligase at the seedling stage and in developing seeds results in increased salt-stress tolerance and elevated fatty acid levels, respectively. This novel approach can help to improve specific traits in crop plants to maintain sustainable agriculture.

## 1. Introduction

With predicted global climate changes, increasing human population, and ongoing agricultural land loss [1,2,3], it is critical to develop novel strategies to generate more resilient crop plants with elevated yields to facilitate sustainable agriculture in the future.

A highly conserved eukaryotic pathway that is involved in various developmental processes, as well as biotic and abiotic stress responses in plants, is the ubiquitin proteasome pathway (UPP) [4]. The pathway is highly flexible and allows plants to react to external signals within minutes, and because of its broad implication on plant physiology and development, it is a good target for bioengineering approaches to modify relevant agricultural traits in commonly grown crop plants [5,6].

The UPP depends on the sequential activities of an E1 ubiquitin (UBQ) activating enzyme, which binds the UBQ and transfers it to a UBQ-conjugating E2 enzyme [7]. The E2 interacts subsequently with an UBQ E3 ligase, an enzyme that facilitates transfer of UBQ moieties to specific protein substrates [7]. Upon building up a UBQ chain, the ubiquitylated protein can be recognized and degraded by the 26S proteasome [7,8,9,10].

E3 ligases are the bottlenecks within the pathway that confer substrate specificity to the UPP. Although they can function as monomeric proteins, most plant E3 ligases form multimeric protein complexes with a cullin as their scaffolding subunit [6,11]. One such complex is the CUL3-RING (CRL3) E3 ligase with a CULLIN3 (CUL3) that binds in its C-terminal region to a RING-finger protein RBX1, and in its N-terminal region to a substrate receptor containing a Broad Complex, Tramtrack, Bric-A-Brac/Pox Virus, And Zinc Finger (BTB/POZ) fold [12,13]. The BTB/POZ fold is required to assemble with CUL3, while a secondary domain is needed to bind to substrate proteins [14,15]. One such secondary domain is Meprin And Traf [Tumor Necrosis Factor Receptor-Associated Factor] Homolog (MATH) [15,16], and the corresponding proteins are referred to as either BTB/POZ-MATH (BPM) or MATH-BTB/POZ (MAB) [12,14].

BPMs are intriguing proteins because they target a surprisingly large and diverse number of substrates as part of a CRL3 E3 ligase complex. So far members from four different transcription factor families, Apetala2/Ethylene Responsive Factor (AP2/ERF) [17,18,19,20], Myelocytomatosis (MYC) [21], Myeloblastosis (MYB) [22,23], and Homeobox-Leucine Zipper (HB) [24], have been described, which connect BPMs with seed oil biosynthesis [17], flowering time control [22,23], abscisic acid (ABA), and jasmonic acid (JA) signaling [21,24], as well as abiotic stress responses [19,20,22,24,25]. BPMs also affect stability of clade A protein phosphatases type 2C (PP2Cs) [26], which function as central negative regulators of an ABA response, and they interact with translation initiation factors [27]. In addition, some BPMs also appear to function independently of a CRL3 E3 ligase since, at least for Arabidopsis BPM1, a CUL3-independent role in regulating RNA-dependent DNA methylation has recently been described [28].

Currently two motifs are known that BPMs recognize in their substrates. First, the SPOP-binding consensus (SBC) that was initially identified for the human BPM ortholog SPECKLE-TYPE POX VIRUS AND ZINC FINGER *PROTEIN* (SPOP) [29]. It comprises five amino acids of the order ϕ–π–S–S/T–S/T (S, serine; T, threonine; ϕ, nonpolar; π, polar) [29], and the motif is well conserved among human and plants [20,29]. Second, a PEST motif was identified as a BPM-interacting site [19]. The motif can be enriched in proline (P), glutamate (E), serine (S), and threonine (T), and is classically associated with protein instability [30,31]. Consequently, deletion of either the PEST or SBC motif in BPM substrates has been demonstrated to significantly increase substrate half-lives [19,20,31,32].

We recently showed that constitutive or stress-dependent expression of a PEST motif is effective at improving salt-stress tolerance, and that specifically the constitutive expression resulted in increased seed sizes in transgenic Arabidopsis plants [31]. The principle of the approach is illustrated in Figure 1, where expression of a PEST motif from BrRAP2.4-1 can reduce CRL3^BPM^ E3 ligase activities by interfering with BPM/substrate assembly. Consequently, BPM substrates become more stable and can function longer in the cell (Figure 1a). To reduce detrimental impacts of PEST overexpression on the plant, we fused the PEST motif with a UBQ and an instability motif enriched in lysines (eK) [31]. This synthetic protein is further referred to as U:PEST. Expression in planta results in a short-lived protein that is quickly degraded via the N-degron pathway [31,33,34], and thereby only transiently interferes with CRL3^BPM^ activities, mainly dependent on the nature of the promoter driving the UBQ:*eK*:*PEST* gene (Figure 1b–d).

These findings prompted investigation into how results from Arabidopsis translate into one of the major oilseed crops, *Brassica napus*, which is considered to be the product of hybridization between *B. rapa* and *B. oleracea* [35], and the low erucic acid variety of which is often referred to as Canola. We were interested in testing overexpression of our U:PEST protein in a seed-specific manner to have as little impact as possible on overall plant development.

Here we describe transient expression of an unstable, synthetic PEST-containing protein that is recognized by BPMs in *Brassica napus* cv. Westar (Canola) to interfere with CRL3^BPM^ activities, and to increase BPM substrate stabilities. The corresponding gene was put under the control of a *Brassica rapa Wrinkled1.2* promoter, which drives expression early in seedling germination and in early seed development. Plants expressing the construct display increased salt-stress tolerance at the seedling stage and increased seed oil biosynthesis. The results shown here are discussed in context with using this approach to transiently block CRL3^BPM^ activities for improving stress tolerance and yield in *Brassica* species, and potentially other crop plants as well.

## 2. Results

### 2.1. Identification of a Suitable Promoter for U:PEST Expression

To identify a *Brassica* gene that is largely expressed specifically in the seed, we searched for *Brassica* orthologs of Arabidopsis *AtWRI1* (*At3g54320*), a gene that has been demonstrated to be primarily expressed in seeds [36]. Since we had the diploid *Brassica rapa* cultivar R-o-18 in the lab, we performed a BLAST search for orthologs at the EnsemblPlants webpage (https://plants.ensembl.org/ (last accessed on 20 October 2022)) and the R-0-18 proteome, and found two proteins with a comparably high homology to AtWRI1 that we named BrWRI1.1 (A09p060030.1_BraROA.1/Brara.I03702.1.p; 81.37% identical to AtWRI1) and BrWRI1.2 (A07p028150.1_BraROA.1/ Brara.G01628.1.p; 81.28% identical to AtWRI1), (Appendix A). Of these, *BrWRI1.2* showed a higher expression in seeds compared to other tissues tested (Figure 2a). To verify that the gene is indeed expressed in seeds, the corresponding promoter was amplified from genomic R-o-18 DNA and put in front of a GUS reporter.

We followed a hypocotyl-based transformation protocol [34] and generated four transgenic *B. napus* cv. Westar (Canola) lines with consistent GUS staining patterns. GUS activity was present in developing seeds (Figure 2i), while no expression was found in either stem or leaf tissues (Figure 2a,j–n). While there was no detectable GUS activity in roots (Figure 2j), RT–qPCR detected some level of *BrWRI1.2* expression in this organ (Figure 2a). We also found some expression, within the first two days of early seedling development, in the root tip and cotyledons after the radicle had emerged from the seed coat (Figure 2b,c), as well as in in the sepals of developing flowers. In addition, older flowers also showed staining in filaments and in the connective tissue between the two thecae (Figure 2e–h). In quintessence, we concluded that the promoter was suitable for a more seed-specific expression of the U:PEST protein, but that we may also observe impacts of the U:PEST protein early in germination and in flower development.

### 2.2. Generation of Transgenic B. napus Plants

For U:PEST expression in *B. napus* cv. Westar plants, an expression construct was built that carried the *proBrWRI1.2* promoter followed by a UBQ:*eK*:*PEST* synthetic gene in the binary vector *pMDC43* [35]. We were able to generate 11 independent T_0_ *proBrWRI1.2*:U:PEST lines. These lines were first selected on hygromycin (hyg) and tested positive for presence of the T-DNA’s *hyg phosphotransferase* gene via PCR on isolated genomic DNA. However, three lines did not produce seeds, and two lines did not show hyg resistance in the T_1_ generation. The remaining six lines were further investigated for expression of the U:PEST construct in 2-day-old seedlings and in flowers, and we found good expression in the U:PEST lines #4, #7, and #8. The expression was based on RT–qPCR and specifically amplification of the *eK*:*PEST* region. These plants were further propagated to the T_3_ generation for stable and non-segregating lines, based on hyg-resistance and robust U:PEST expression (Figure 3). Of interest is that while U:PEST #7 and #8 were comparably consistently expressed in 2-day-old seedlings and mature flowers, U:PEST #4 was strongly expressed in the seedlings compared to flowers (Figure 3). The T_3_ generation was then taken for detailed phenotypic analysis; the three U:PEST lines were in general indistinguishable from the *B. napus* cv. Westar wild-type (WT) background plants with respect to growth, and we did not observe any obvious changes in flowering time or flower development (see Appendix A for flower phenotype and fully grown habitus).

### 2.3. proBrWRI1.2:U:PEST Seedlings Have Increased Salt-Stress Tolerance

We first tested plants for salt tolerance focusing on germination and early seedling development as the *BrWRI1.2* promoter was mainly active within the first two to three days after radicle emergence from the seed coat, based on GUS data (Figure 2a,c). For these experiments seeds were plated on basic growth medium with or without supplemental NaCl (150 mM). We observed a mild variation among the different lines and WT on the control plates (no salt) with respect to the germination frequency, but all three U:PEST lines and the WT fully germinated within the first week (Figure 4a). U:PEST #8 germinated slightly faster, while U:PESTs #4 and #7 were slightly slower compared to wild type (Figure 4a and Appendix A). However, none of the lines germinated significantly differently than wild type.

Interestingly, U:PEST #4, #7, and #8 all germinated significantly faster and to a higher final number than wild type seeds when exposed to salt stress (Figure 4b and Appendix A). Within six days the three transgenic lines and WT reached a plateau, indicating that the maximum germination rate had been reached. WT seeds barely germinated and became stagnant at less than 5% of the total seeds plated on medium containing 150 mM NaCl on average, demonstrating high salt sensitivity in Westar WT. In comparison, 80% of seeds from U:PEST lines #7 and #8 germinated after six days, while germination of seeds from U:PEST line #4 stagnated at 20% (Figure 4b and Appendix A). However, all lines were clearly more salt tolerant than WT seeds, signifying that the U:PEST construct can very efficiently improve plant performance when exposed to this stressor.

### 2.4. proBrWRI1.2:U:PEST Plants Have Larger Seeds and Increased Fatty Acid Content

We analyzed seed phenotypes for the three U:PEST lines, and looked first at a simple size comparison to WT. All our transgenic lines clearly had larger seeds than WT (shown in Figure 5a for U:PEST #7), and this was further confirmed by weight data. On average seed weights were increased by 14% (U:PEST #8), 18% (U:PEST #4), and 25% (U:PEST #7) compared to WT (Figure 5b). We also investigated embryo size and could see a similar trend as observed for the overall seed weight (Figure 5c). Whole-seed fatty acid analysis for U:PEST #7 and #8 showed significant increases in total fatty acid methyl ester (FAME) content per seed with average increases of 28% and 29%, respectively (Figure 5d). For U:PEST #4 we also detected an average increase of 13%, but variation among the samples was too high to estimate a significant increase compared to WT (Figure 5d). Calculation of μg FAME per mg seed did not yield any significant difference to WT, which indicates that any changes measured in total fatty acid content were proportional to the overall changes in seed weight (Figure 5e). Profiling of the different classes of seed fatty acids (FA) showed general significant increases in the major Westar variety FAs including palmitic acid (16:0), stearic acid (18:0), oleic acid (18:1), linoleic acid (18:2), and α-linolenic acid (18:3) for U:PEST #7 and #8 compared to WT (Figure 5e). U:PEST #4 showed only significant increases in the stearic acid fraction (Figure 5f). In contrast, most of the minor very long chain FAs ranging from arachidic acid (20:0), 11-eicosenoic acid (20:1), behenic acid (22:0), to lignoceric acid (24:0) were not significantly changed in the different U:PEST lines (Figure 5f). Only arachidic acid was significantly increased in U:PEST #4, and 11-eicosenoic acid in U:PEST #8 (Figure 5f).

## 3. Discussion

Here we show that transient expression of an unstable U:PEST protein under the control of a *BrWRI1.2* promoter specifically impacts salt-stress tolerance in germinating seedlings and accumulation of storage oils in developing seeds.

The overall activity of the cloned *BrWRI1.2* promoter is mostly in agreement with the RT–qPCR results, except for roots, where no GUS staining was detectable in older plants (though some activity was detectable early in germinating seedlings). The reasons for this discrepancy are unclear, but may be located, for example, in species-specific differences between *B. rapa* and *B. napus*, or it might be that the cloned promoter simply lacks elements required for root-specific expression. However, for this work it was a very suitable tool to effectively control U:PEST expression at defined developmental stages. Accordingly, we did not observe any obvious changes in general development between the three U:PEST lines and WT. We did not see that sepal and filament growth were affected despite GUS evidence of *BrWRI1.2* activity in those tissues, which could indicate either that CRL3^BPM^s do not play critical roles in these tissues, or that inhibition by the U:PEST protein was not strong enough to generate visible changes.

The increased salt-stress tolerance of the U:PEST lines is in agreement with our previous work in Arabidopsis, where plants expressing the U:PEST protein, either constitutively or under the control of a *RD29a* promoter, showed consistently reduced NaCl sensitivity from the early germination stage until adulthood [31]. Because of the nature of the restricted activities of the *BrWRI1.2* promoter, we did not test later developmental stages for salt tolerance, since within the first three days no GUS activity was detectable in the growing plants until the point of flower development. However, based on results from Arabidopsis one may expect that the U:PEST expression facilitates salt tolerance not only in early germination, but also in mature *B. napus* plants [31]. It was unexpected that the U:PEST expression in seedlings was highest in line #4, which had the lowest tolerance to salt. We currently do not have a clear explanation for this, but based on the lower NaCl tolerance compared to U:PEST #7 and #8, it is possible that RT–qPCR data do not reflect the translation rate in U:PEST #4. As previously reported, the U:PEST protein is barely detectable, even when expressed under the control of a *35S* promoter, likely due to its high instability [31]. The *proBrWRI1.2* is comparably weak, and we were not able to detect the protein in our *B. napus* plants, so we are unable to compare protein differences among the three transgenic lines.

Analysis of seeds showed that the U:PEST seed weights and fatty acid contents were significantly increased in comparison to WT. These findings are also in agreement with previous findings from Arabidopsis, where seeds were significantly heavier and larger when CRL3^BPM^ activities were either reduced by using an artificial microRNA (amiRNA) against BPMs or the expression of the U:PEST construct [17,31]. The increase in total fatty acid content in seeds was lower than previously accomplished using the amiRNA approach [17], but was similar to earlier reports where constitutive overexpression of a *WRI1* gene was attempted [37,38,39,40]. However, the overall impact of the amiRNA resulted in strong developmental changes, especially in smaller and slower growing plants, and an aberrant flowering time [17,23], something we did not observe in this study. It is therefore likely that the increase in average seed weight of up to 29% in the U:PEST plants is actually much more efficient when compared to the previous amiRNA approach, since the U:PEST plants exhibited an overall phenotype similar to WT plants. This will require a more detailed analysis of total seeds per plant, and a generation of a harvest index. However, preliminary data show that the average seed number per silique is not different in the U:PEST lines when compared to WT (Appendix A).

Of note is that the overall composition of FAs was not significantly altered. This was expectable, since the increased FA content is likely the result of a more stable WRI1 protein in the U:PEST plants [17,32,39]. WRI1 has been established as a master regulator of FA biosynthesis in plants that is required for the overall rate of storage oil production, but not for fatty acid composition [39]. Our results also showed that the FA composition per seed weight, and thus quality in the U:PEST plants, is in principle comparable to the WT plants. The increased FA amounts per seed indicate that U:PEST plants have a higher caloric content and can have a significantly higher yield on the same acreage compared to WT plants.

The larger seeds are not just caused by the WRI1 activities but also by other proteins, as, for example, overexpression of MYB56, another CRL3^BPM^ substrate, has been reported to result in larger embryos [41]. However, as of now it is only poorly understood how CRL3^BPM^ affects seed development on a system-wide level. It will therefore be of interest to understand the protein activities impacted by this E3 ligase more globally in the embryo and seeds in the future.

Based on the current findings, it is of interest to engineer double transformants. For example, plants may express the U:PEST under a strictly stress-inducible promoter, while a second construct drives U:PEST expression under the control of a stronger and more seed-specific promoter than the *proBrWRI1.2*, such as a *beta-conglycinin* promoter [42]. This may yield plants that have a robust abiotic stress tolerance throughout development, with the capacity to also produce larger seeds with significantly increased FA contents, beyond what is shown in this work.

## 4. Materials & Methods

### 4.1. Plant Growth, Transformation, and Phenotypic Analysis

*Brassica napus* cv. Westar (Canola) and *B. rapa* cv. R-0-18 plants were grown under long-day (16 h light: 8 h dark; light intensity growth chamber 91.8 μmol/m^2^/s in (Percival, IA); light intensity greenhouse 231.8 μmol/m^2^/s) and standard growth conditions (20 °C, ~50% humidity) in soil and sterile culture (basic ATS medium without supplemental sucrose [43]). *B. napus* plant transformations were performed using an agrobacterium, hypocotyl-based protocol modified from [44]. For germination assays only seeds from plants grown at the same time and under the same conditions were used, and seeds were plated on basic ATS medium supplemented with or without salt (NaCl). The time point when the radicle first emerged from the seed coat was recorded as the end of germination and referred to as ‘final germination’. Seed weight was measured on a Denver analytical scale (Denver Instrument, Bohemia, NY, USA). For embryo-size analysis, seeds were imbibed in water for 24 h. Before embryos were carefully squeezed out of the seed with a blunt forceps, a small slit was made into the seed coat with a razor blade. Embryo size analysis based on area was performed from jpeg images using the ImageJ image processing package Fiji (https://imagej.net/software/fiji/ (accessed on 17 February 2023)).

### 4.2. Generation of Expression Constructs

For the *BrWRI1.2* promoter, a 1500 bp long fragment upstream of the *BrWRI1.2* (*Bra003178*) coding region was amplified from *B. rapa* R-0-18 genomic DNA. The fragment was cloned via a gateway BP reaction into *pDONR221* (Thermo Fisher Scientific, Waltham, MA, USA) before being shuffled via a gateway LR reaction (Thermo Fisher Scientific, Waltham, MA, USA) into the binary vector *pMDC162* [45] to allow *proBrWRI1.2*-controlled *GUS* expression. In addition, the same promoter was cloned into the gateway-compatible, binary vector *pMDC43* via *Hind*III/*Kpn*I sites, replacing its *35S* promoter, to generate a *proWRI1.2*-*pMDC43* vector. For the U:PEST construct, a PEST-containing C-terminal region from *BrRAP2.4-1* (*Bra003659*) was amplified from the corresponding cDNA and fused to a UBQ:*eK* element as described in [31]. The corresponding U:PEST was first cloned into *pDONR221* before being shuffled into *proWRI1.2-pMDC43* using gateway BP and LR reactions (Thermo Fisher Scientific, Waltham, MA, USA), respectively. For a full sequence of the UBQ:*eK*:*PEST* see [31]. All primers used are listed in Appendix A All constructs were verified before usage by sequencing for correct translational frame and absence of PCR-generated mutations.

### 4.3. GUS Staining

GUS staining was performed according to [46]. Seeds and embryos were stained in X-Gluc (5-bromo-4-chloro-3-indolyl-ß-D-glucuronide; Gold Biotechnology, MO) solution (1 mM X-Gluc, 50 mM Na-phosphate buffer, pH 7.0) for 6 h at 37 °C, after which staining was stopped by washing tissues twice in 1 mL 70% ethanol.

### 4.4. mRNA Isolation and RT–qPCR Analysis

Total RNA was isolated using an Isolate RNA Minikit (Bioline, Wayne, NJ, USA). The RNA was quantified with a NanoDrop ND-1000 spectrophotometer (ThermoScientific Fisher, Waltham, MA, USA), and 100 ng per RT–qPCR reaction was directly added to an AzuraQuant 1-Step qPCR Mix (Azura Genomics Inc., Raynham, MA, USA), which allows for reverse transcription and real-time PCR amplification in a single tube. Reactions were performed in a 7500 Fast Real-Time PCR system (Applied Biosystems, Foster City, CA, USA) as described earlier [19]. *B. napus Actin* (*AF111812*; [47]) mRNA was used as the internal control. For in planta detection of U:PEST expression specific primer pairs were designed that covered the eK and PEST sequences. All experiments were repeated at least three times as biological replicates, if not otherwise stated. The 2(-delta delta C(T)) method was used to calculate relative gene expression [48]. All primers were ordered from MilliporeSigma, Bedford, MA, USA, and sequences are listed in Appendix A.

### 4.5. Fatty Acid Analysis

Seed oil content in *Brassica napus* cv. Westar (Canola) was analyzed from 10 seeds of individual plants that were dried over silica beads (Thermo Fisher Scientific, Waltham, MA, USA). Seeds were weighed and combined in a 10 cm × 1 cm glass tube with 1 mL of toluene containing 80 µg tripentadecanoin (15:0) as an internal standard and 0.0005% (*w*/*v*) butylated hydroxy toluene (BHT). Samples were then ground using a tissue homogenizer (Polytron PT 2500E, Kinematica, Bohemia, NY, USA) and seed material was rinsed from the homogenizer with an additional 2 mL methanol that was combined with the sample. Next, 1 mL of 15% (v/v) conc. sulfuric acid in MeOH was added and tubes were sealed using PTFE lined caps. Samples were incubated at 85–90 °C for 2 h with samples being inverted every half hour. After samples cooled, 2 mL of hexane and 1.5 mL 0.88% (*w*/*v*) potassium chloride (KCl) were added, samples were vortexed, and phase separation was achieved by centrifuging samples at 3000× *g* for 3 min. Fatty acid methyl esters (FAME) extracted in the upper hexane layer were analyzed using Agilent model 7890 GC with an attached flame ionization detector and a DB FATWAX UI column (30 m × 0.25 µm; Agilent, Santa Clara, CA, USA) following the method described in [49].

### 4.6. Imaging

Pictures were taken with a Nikon Z7II and Nikkor Z MC 105 mm f/2.8 VR S lens (Nikon, Japan).

### 4.7. Statistical Analysis

For statistical analysis Student’s t-tests (heteroscedastic, two-tailed distribution) were performed using Microsoft Excel software. Values with *p* < 0.05 were considered significant. Error bars show standard deviations. If not otherwise stated, all calculated data are based on at least three biological replicates.

## 5. Conclusions

In conclusion, expression of U:PEST under the control of specific promoters has been proven to be quite powerful in impacting certain plant traits without affecting general development ([31]; this work). The *BrWRI1.2* promoter is an especially powerful tool to affect different growth stages such as germination and early seedling growth, as well as seed development in a single approach. Because CRL3^BPM^ E3 ligases are highly conserved among plants, this work may also have impacts on developing new strategies to improve stress resilience and yield in crop plants. It has, therefore, high relevance for sustainable agriculture in the context of global climate changes and increasing frequencies of sub-optimal growth conditions.

## Figures and Tables

**Figure 1 plants-12-01085-f001:**
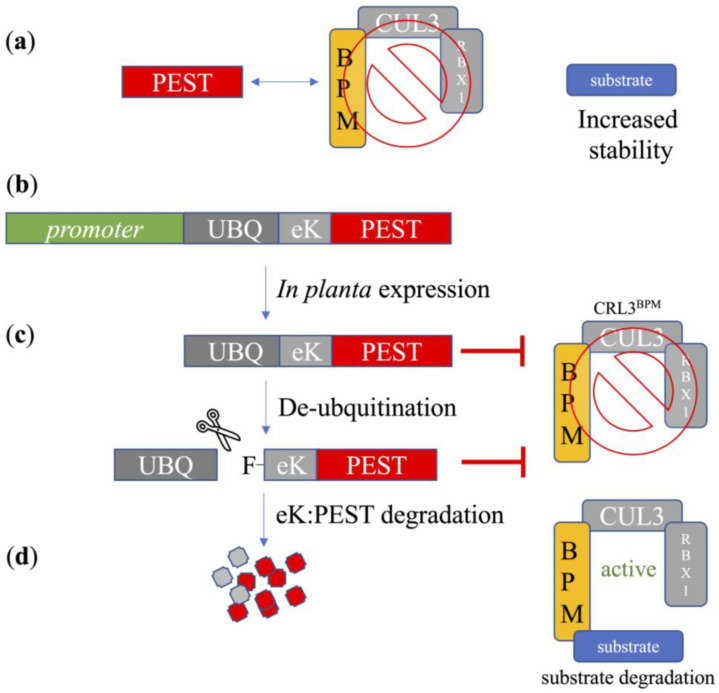
Schematic drawing of the transient interference of CRL3^BPM^ activities. (**a**) Presence of a PEST motif can reduce CRL3^BPM^ activity by occupying BPM substrate binding sites. This occupation is anticipated to increase the half-lives and activities of BPM substrates. (**b**) Expression of the UBQ:*eK*:*PEST* construct under the control of a specific promoter in planta leads to reduced CRL3^BPM^ activities whenever the given promoter is active. (**c**) Cleavage of the UBQ moiety by endogenous de-ubiquitination enzymes (represented by the scissor symbol) results in exposure of a phenylalanine (F) residue. This targets the remaining *eK*:*PEST* for degradation via the N-degron pathway. (**d**) Degradation of *eK*:*PEST* releases BPMs and allows CRL3^BPM^ E3 ligases to normally target their substrate proteins again.

**Figure 2 plants-12-01085-f002:**
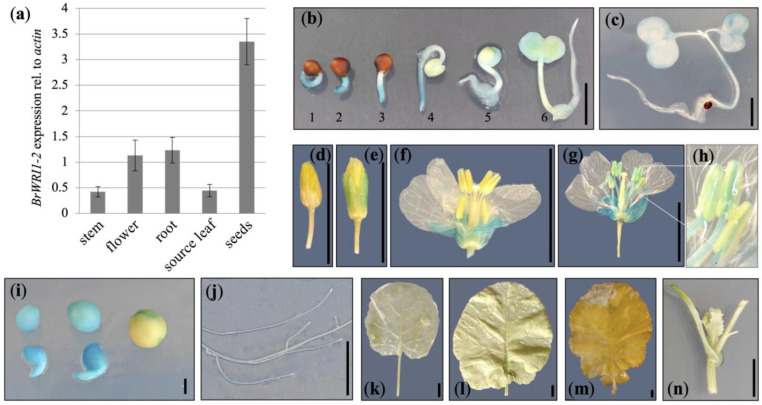
Expression analysis of a *B. rapa WRI1.2* promoter. (**a**) RT–qPCR in different *B. rapa* R-o-18 tissues shows *BrWRI1.2* expression predominantly in seeds, but also in flowers and roots. (**b**–**n**) *GUS* expression under the control of the *BrWRI1.2* promoter in different *B. napus* tissues. (**b**) Early seedling growth stages (1, 20 h; 2, 25 h; 3, 30 h; 4, 45 h; 5, 70 h; all times refer to post plating). (**c**) A 3-day-old seedling. (**d**–**g**) Flowers at different developmental stages. (**h**) Enlarged anthers from (**g**). (**i**) Developing seeds. (**j**) Root. (**k**) Sink leaf. (**l**) Source leaf. (**m**) Senescent leaf. (**n**) Stem with budding leaf. The scale bar lengths are 1 mm in (**i**), 5 mm in (**b**,**c**), and 10 mm in (**d**–**g**,**j**–**n**).

**Figure 3 plants-12-01085-f003:**
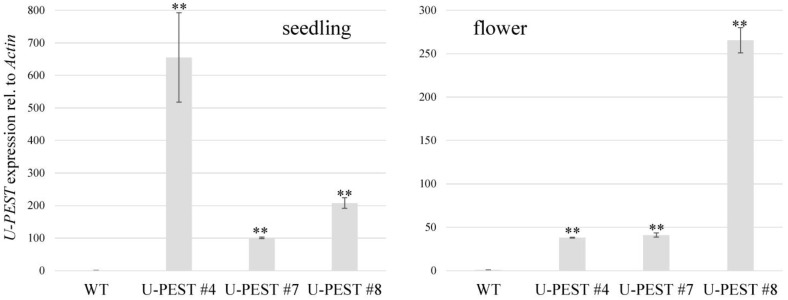
RT–qPCR results for germinating seedlings (2 days old; **left**) and fully developed flowers (**right**). Data are based on three biological replicates from three different plants. **, *p* < 0.01 compared to WT.

**Figure 4 plants-12-01085-f004:**
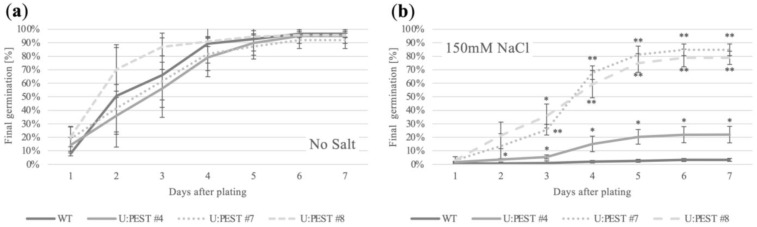
U:PEST lines display increased salt tolerance compared to WT plants. (**a**) Germination on sterile medium plates without supplemental NaCl shows no significant differences between WT and the three U:PEST lines. (**b**) Presence of 150 mM NaCl severely inhibits germination of WT seeds while all three U:PEST lines germinate faster and to a higher degree. Student’s *t*-test; *, *p* < 0.05; **, *p* < 0.01. 20 seeds from three different plants/genetic backgrounds were used and tested (*n* = 3) independently.

**Figure 5 plants-12-01085-f005:**
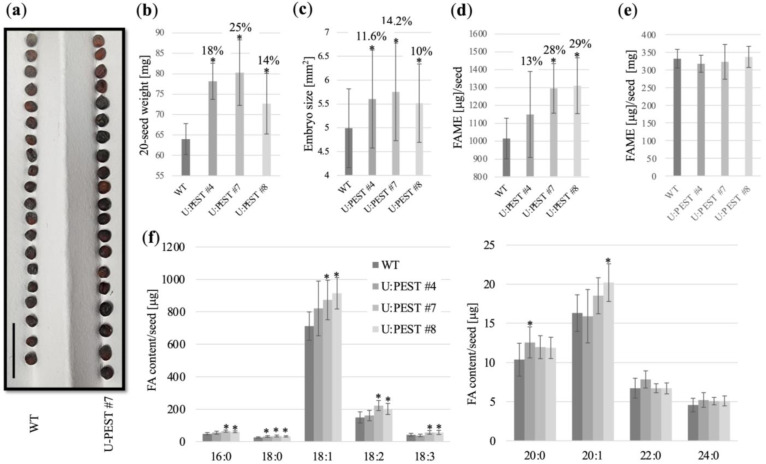
Seed size, weight, and fatty acid analysis. (**a**) Comparison of 20 WT and U:PEST #7 seeds. Scale bar = 10 mm. (**b**) Seed weight measurements. (**c**) Embryo size measurements. Data are based on 15 embryos from three independent plants per genetic background. (**d**) Total fatty acid methyl ester (FAME) measurements in WT and U:PEST #4, #7, and #8 lines. (**e**) Changes in FAME contents are proportional to seed weight. (**f**) Quantification of individual fatty acid (FA) contents in WT and the U:PEST lines. Student’s *t*-test; *, *p* < 0.05. All data (**b**–**f**) are based on three biological replicates from three different plants per genetic background.

## Data Availability

No other new data were created or analyzed in this study than already shown. Data sharing is therefore not applicable to this article.

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
