# Peer review of "Brassica napus Plants Gain Improved Salt-Stress Tolerance and Increased Storage Oil Biosynthesis by Interfering with CRL3BPM Activities"

_plants, 2023, doi:10.3390/plants12051085_

Round 1

Reviewer 1 Report

Brassica napus plants gain improved salt stress tolerance and increased storage oil biosynthesis by interfering with CRCBPM activities.

This manuscript was written the generating new strategy in increased salt stress tolerance and elevated fatty acid levels. It will be a benefit to maintain sustainable agriculture. In use, seed size, weight, and fatty acid composition variations are important trails in B. napus. 

There are the serval mistakes in text with too many using ‘an adverbial phrase’ and in be mixed on ‘result’ ad ‘discussion’. You should keep the simple expression follows results. Also, all figure images should be visible by using bold or ‘Arial font’.

In experiment figure 4, U-PEST lines display increased salt tolerance compared to WT plants without NaCl germination efficiency is poor that all germination reached 80% on fifth day. Figure 4 experiment displays on controversial data, comparing Figure 3, seedling. U-PEST expression rel. to Actin showed the highest expression in lne#4 in Figure 3, but the germination in NaCl displayed the lowest efficiency. Why??

Line 143: Delete ‘B. rapa’. BrRAP2.4-1 includes Brassica rapa. 

Line 156: Delete ‘(b)’

Line 159: Delete ‘(F)’.

Line 161: Delete ‘again’.

Line 162-167: Move to ‘discussion’

Line 168: Move to ‘discussion’

Line 180-182: RT-qPCR in different B. rapa tissues shows BrWRI1.2 expression……..B. napus tissues (a-n) Which tissues are right?

Line 183-188: The scale bar indicated in one sentence. 

* The scale bar lengths are 5mm in (b, c) and is 1mm in (i), and others are 10mm in…..

Line: 190-191: Delete ‘As anticipated’.

Line 192: Delete ‘indeed’.

Line 190-200: Authors wrote ‘an adverbial phrase’ in many times such as ‘as anticipated’, ‘indeed’, ‘while’ two times. ‘however’, ‘also’, ‘as well as’. ‘in addition’, and ‘in quintessence’.

Author Response

Responds: Dear Reviewer, first of all, thank you for the work you put into reviewing our manuscript. In the following we went step-by-step through the different points. For your information, we added a new sub-figure to Fig. 5 showing embryo size, and we also added a new Table S2 with additional analyzes of the shown germination data in Fig. 4.

Comments and Suggestions for Authors

Brassica napus plants gain improved salt stress tolerance and increased storage oil biosynthesis by interfering with CRCBPM activities.

This manuscript was written the generating new strategy in increased salt stress tolerance and elevated fatty acid levels. It will be a benefit to maintain sustainable agriculture. In use, seed size, weight, and fatty acid composition variations are important trails in B. napus. 

There are the serval mistakes in text with too many using ‘an adverbial phrase’ and in be mixed on ‘result’ ad ‘discussion’. You should keep the simple expression follows results. Also, all figure images should be visible by using bold or ‘Arial font’.

>Responds: Thank you for the comment. This is a very generic statement in the first place (…There are the serval mistakes…). Also, MDPI has no guidelines related to ‘adverbial phrases’, and we consider the writing appropriate. We disagree with the font. Times New Roman works fine and we have made all lettering already in bold.

In experiment figure 4, U-PEST lines display increased salt tolerance compared to WT plants without NaCl germination efficiency is poor that all germination reached 80% on fifth day. Figure 4 experiment displays on controversial data, comparing Figure 3, seedling. U-PEST expression rel. to Actin showed the highest expression in lne#4 in Figure 3, but the germination in NaCl displayed the lowest efficiency. Why??

>Responds: Thank you for the question and good point. We agree that this is unexpected, and cannot provide a precise answer to that. It is likely that the transcription levels do not reflect the protein level, which would not be uncommon. We tried to detect protein but were not able to get satisfying results. This was the same for our work in Arabidopsis, which even used a very strong 35S promoter (in comparison to the BrWRI1.2 promoter used in this work). The U-PEST protein is designed to be very unstable and this is likely the reason for detection difficulties. In responds to the reviewer’s criticism we added a section to the discussion (lines 265-275) to address this.

Line 143: Delete ‘B. rapa’. BrRAP2.4-1 includes Brassica rapa. 

>Responds: has been deleted

Line 156: Delete ‘(b)’

>Responds: has been deleted

Line 159: Delete ‘(F)’.

Responds: we like to keep this here because the (F) is the abbreviation for phenylalanine and explanation for the F in Fig. 1c

>Line 161: Delete ‘again’.

Responds: we would like to keep this, as the ‘again’ in the legend of Fig. 1 (l. 80) emphasizes that CRL3BPM returns to its normal activities.

Line 162-167: Move to ‘discussion’

>responds: we prefer to leave it here

Line 168: Move to ‘discussion’

>responds: we prefer to leave it here

Line 180-182: RT-qPCR in different B. rapa tissues shows BrWRI1.2 expression……..B. napus tissues (a-n) Which tissues are right?

>responds: thank you for the comment. It needs to be (b-n). This was changed, and we moved this before ‘GUS expression’ now. We also added to Figure (a) R-o-18 to emphasize that the same B. rapa was used for promoter identification as well as for RT-qPCR analysis.

Line 183-188: The scale bar indicated in one sentence. 

* The scale bar lengths are 5mm in (b, c) and is 1mm in (i), and others are 10mm in…..

>responds: this has been bundled now into one sentence at the very end of the legend.

Line: 190-191: Delete ‘As anticipated’.

>Responds: has been deleted

Line 192: Delete ‘indeed’.

>Responds: has been deleted

Line 190-200: Authors wrote ‘an adverbial phrase’ in many times such as ‘as anticipated’, ‘indeed’, ‘while’ two times. ‘however’, ‘also’, ‘as well as’. ‘in addition’, and ‘in quintessence’.

>Responds: it is not clear what the reviewer is suggesting here. The text reads fine, the meaning is clear, and MDPI has no guidelines for ‘adverbial phrases’ that would restrict usage. We would therefore like to keep as is.

Reviewer 2 Report

The manuscript entitled “Brassica napus plants gain improved salt stress tolerance and 2 increased storage oil biosynthesis by interfering with CRL3BPM activities” by Emily Corbridge, Alexandra MacGregor, Raed Al-Saharin, Matthew G. Garneau, Samuel Smalley, Sutton Mooney, Sanja Roje, Philip D. Bates, and Hanjo Hellmann tackles the problem of salt tolerance in Brassica napus. I found the manuscript important from both scientific and economical points of view. Salinity is one of the major environmental stresses affecting several crops. Rapeseed is one of the most important oil plants grown worldwide. In general, the manuscript is quite well-written however authors should work out the details. For example, it is not clear why the authors decided to analyze the plant under salt stress. Did the authors consider other types of stress? If not then why? Why authors decided to use promoter specific for seeds? I do not agree that using seed-specific promoter would limit the negative impact on overall plant development. Germination is a crucial process in the plant life cycle therefore everything that is happening during germination might have an impact on subsequent growth stages. I think that the manuscript needs to be rewritten to clearly explain the author’s motivation for planning the experiments in such a way.   

Specific comments:

1. Abstract is too generic. Please provide more details.

2. Abstract:

a. Why names of proteins/domains are written in capital letters?

b. Please provide one main aim of this study.

c. It is not clear what is the novelty of the manuscript.

3. Materials and methods:

a. Page 2, line 82: Please provide the light intensity.

b. Page 2, lines 87-88: When the radicle emerges from the seed coat is not a beginning of germination, it is a moment when the germination is completed.

c. Page 3, line 108: Total RNA or mRNA was isolated? How was the quantity and integrity of RNA analyzed?

4. Results:

a. Pages 3-4, lines 139-167, figure 1: this part is not a result of this study. I do understand that this is presented to explain this study but it should be moved to the introduction or could be presented before the results if results and discussion are combined.

b. Figure 2o: It is unclear what authors wanted to present. If the x-axis is a time then seedlings, flowers, and developing seeds are in the same time point represented by the red arrow? Please modify this figure however I do not think it is necessary to include it in the manuscript.

c. Figure 4: The quality of this figure should be improved. Authors should consider replacing this figure with a table showing not only the germinability by also other germination-related parameters including germination index, coefficient of germination velocity, mean germination time, etc.

d. Figure 5: The quality of this figure should be improved.  

5. Discussion lacks a more general picture. All results are discussed separately and there is no general conclusion provided. In the conclusion section plans for further experiments are presented instead of proper conclusions. I agree that is important to show how the research will be developed in the future however clear conclusions from the results presented in the manuscript are mandatory.

Author Response

Comments and Suggestions for Authors

The manuscript entitled “Brassica napus plants gain improved salt stress tolerance and 2 increased storage oil biosynthesis by interfering with CRL3BPM activities” by Emily Corbridge, Alexandra MacGregor, Raed Al-Saharin, Matthew G. Garneau, Samuel Smalley, Sutton Mooney, Sanja Roje, Philip D. Bates, and Hanjo Hellmann tackles the problem of salt tolerance in Brassica napus. I found the manuscript important from both scientific and economical points of view. Salinity is one of the major environmental stresses affecting several crops. Rapeseed is one of the most important oil plants grown worldwide. In general, the manuscript is quite well-written however authors should work out the details. For example, it is not clear why the authors decided to analyze the plant under salt stress. Did the authors consider other types of stress? If not then why? Why authors decided to use promoter specific for seeds? I do not agree that using seed-specific promoter would limit the negative impact on overall plant development. Germination is a crucial process in the plant life cycle therefore everything that is happening during germination might have an impact on subsequent growth stages. I think that the manuscript needs to be rewritten to clearly explain the author’s motivation for planning the experiments in such a way.   

>Responds: Dear Reviewer, first of all thank you for the work you put into reviewing our manuscript. I guess the point of the reviewer about tissue and organ-specific promoter usages is discussable. I would fully agree that the germination stage is crucial. Of course, if germination doesn’t work not much else will. But I would strongly disagree that everything later on is affected by this developmental stage. One beauty about plants is their open body architecture (in contrast to animals) and their flexibility to develop new organs throughout their life cycle. The new developments are not all defined at the germination stage. Anyway, the point of using a tissue-specific promoter is to minimize impacts on other developmental stages. This is accurately stated in the manuscript (why we were interested in a seed-specific promoter e.g., copied from the manuscript “We were interested in testing overexpression of our U:PEST protein in a seed-specific manner to have as little impact as possible on overall plant development.”), that is what we observe (seed and seedling-specific impacts), and I would consider it therefore a good strategy, and that there is no need to re-write the manuscript! As a general note we added a new sub-figure to Fig. 5 that shows embryo size, and we also added a Table S2 with a additional analyzes of germination data related to Fig. 4.

Specific comments:

  1. Abstract is too generic. Please provide more details.

>responds: the abstract is to our feeling concise and accurate as it describes the approach and outcomes of the research without getting lost in too much detail.

  1. Abstract:
  2. Why names of proteins/domains are written in capital letters?

>responds: we have always provided domain names in capital letters as this emphasizes the letters used for the abbreviation.

  1. Please provide one main aim of this study.
  2. It is not clear what is the novelty of the manuscript.

>Responds: To address the reviewer’s criticism from (b) and (c) we added “The aim of this research was to transiently downregulate an E3 ligase that uses BTB/POZ-MATH proteins as substrate adaptors in a tissue-specific manner.” and “novel” to “This novel approach…” to the abstract.

  1. Materials and methods:
  2. Page 2, line 82: Please provide the light intensity.

>Responds: we added this to the method section 2.1 p. 3 lines 96/97 for growth chamber and greenhouse.

  1. Page 2, lines 87-88: When the radicle emerges from the seed coat is not a beginning of germination, it is a moment when the germination is completed.

>Responds: thank you, that is correct, but it is still a very good optical indicator for the speed of germination. We would keep this wording in this context but have changed the labelling on the graphs to “final germination [%]. We also rephrased the developmental stage of activity of the GUS promoter to “early in seedling development” (p.7 line164), and in the legend of Figure 2 to “early seedling growth stages”. We also changed phrasing on p.9 lines 173/174.

  1. Page 3, line 108: Total RNA or mRNA was isolated? How was the quantity and integrity of RNA analyzed?

>Responds: in the method part 2.4 l. 120 it states ‘total RNA’. Originally ‘total Arabidopsis RNA, but the Arabidopsis is deleted now. RNA was quantified using a ND-1000 nanodrop machine. This has been added to the method part.

  1. Results:
  2. Pages 3-4, lines 139-167, figure 1: this part is not a result of this study. I do understand that this is presented to explain this study but it should be moved to the introduction or could be presented before the results if results and discussion are combined.

>Responds: thank you for the recommendation. We have moved the whole section, including the figure, into the introduction.

  1. Figure 2o: It is unclear what authors wanted to present. If the x-axis is a time then seedlings, flowers, and developing seeds are in the same time point represented by the red arrow? Please modify this figure however I do not think it is necessary to include it in the manuscript.

>Responds:  We took it out.

  1. Figure 4: The quality of this figure should be improved. Authors should consider replacing this figure with a table showing not only the germinability by also other germination-related parameters including germination index, coefficient of germination velocity, mean germination time, etc.

>Responds: We prefer the presentation as is and do not want to replace it. But we have added a table with the requested parameters as a supplementary table. We have exchanged this figure for a higher resolution version.

  1. Figure 5: The quality of this figure should be improved.  

>Responds: We have exchanged this figure with a higher resolution version. We also added another subfigure 5c to show in addition to seed weight also embryo size.

  1. Discussion lacks a more general picture. All results are discussed separately and there is no general conclusion provided. In the conclusion section plans for further experiments are presented instead of proper conclusions. I agree that is important to show how the research will be developed in the future however clear conclusions from the results presented in the manuscript are mandatory.

>responds: We consider the discussion to be adequately focused but moved the engineering portion from the conclusions into the discussion part. In responds to the criticism about our conclusions, we added a section that addresses more broadly the relevance of the shown research for sustainable agriculture in context with increasing frequencies of harsh weather conditions.

Round 2

Reviewer 1 Report

OK